# Exploring Albumin Functionality Assays: A Pilot Study on Sepsis Evaluation in Intensive Care Medicine

**DOI:** 10.3390/ijms241612551

**Published:** 2023-08-08

**Authors:** Gerd Klinkmann, Katja Waterstradt, Sebastian Klammt, Kerstin Schnurr, Jens-Christian Schewe, Reinhold Wasserkort, Steffen Mitzner

**Affiliations:** 1Department of Anaesthesiology, Intensive Care Medicine and Pain Therapy, University Medical Center Rostock, Schillingallee 35, 18057 Rostock, Germany; 2Fraunhofer Institute for Cell Therapy and Immunology, Department of Extracorporeal Therapy Systems, Schillingallee 68, 18057 Rostock, Germany; 3Department of Research and Development, MedInnovation GmbH, 12487 Berlin, Germany; 4Division of Nephrology, Department of Internal Medicine, University Medical Center Rostock, Ernst-Heydemann-Str. 6, 18057 Rostock, Germany

**Keywords:** albumin, binding capacity, electron paramagnetic resonance, spin-probe technique, critical care, intensive care medicine, sepsis, septic shock

## Abstract

Human serum albumin (HSA) as the most abundant plasma protein carries multifunctional properties. A major determinant of the efficacy of albumin relies on its potent binding capacity for toxins and pharmaceutical agents. Albumin binding is impaired in pathological conditions, affecting its function as a molecular scavenger. Limited knowledge is available on the functional properties of albumin in critically ill patients with sepsis or septic shock. A prospective, non-interventional clinical trial assessed blood samples from 26 intensive care patients. Albumin-binding capacity (ABiC) was determined by quantifying the unbound fraction of the fluorescent marker, dansyl sarcosine. Electron paramagnetic resonance fatty acid spin-probe evaluated albumin’s binding and detoxification efficiencies. Binding efficiency (BE) reflects the strength and amount of bound fatty acids, and detoxification efficiency (DTE) indicates the molecular flexibility of patient albumin. ABiC, BE, and DTE effectively differentiated control patients from those with sepsis or septic shock (AUROC > 0.8). The diagnostic performance of BE showed similarities to procalcitonin. Albumin functionality correlates with parameters for inflammation, hepatic, or renal insufficiency. Albumin-binding function was significantly reduced in critically ill patients with sepsis or septic shock. These findings may help develop patient-specific algorithms for new diagnostic and therapeutic approaches.

## 1. Introduction

Sepsis is a critical medical condition characterized by life-threatening dysfunction of organs resulting from an imbalanced host response to infection. Within the spectrum of clinical manifestations of sepsis, septic shock can be identified as a distinct subset where underlying circulatory, cellular, and metabolic abnormalities contribute to a higher risk of mortality compared to sepsis alone [1]. In the year 2017 alone, an estimated 48.9 million cases of sepsis were reported, accompanied by a staggering 11.0 million sepsis-related deaths, accounting for 19.7% of total recorded mortalities worldwide [2]. Notably, the mortality rate attributed to severe septic shock in Europe and North America has been approximated at 38% [1]. However, it is crucial to acknowledge that these figures likely underestimate the true prevalence and mortality burden as numerous cases go unreported [3]. Despite significant progress in therapeutic management, patients afflicted by sepsis and septic shock continue to face a heightened risk of in-hospital mortality, contributing to around 20% of all-cause deaths globally [4,5,6]. This high mortality rate in sepsis results from challenging diagnosis, intricate treatment protocols, the presence of comorbidities, growing antimicrobial resistance, and the critical time-dependent nature of the condition [7]. To effectively address these factors and enhance patient outcomes while mitigating the impact of sepsis, a comprehensive and well-coordinated approach is essential.

Human serum albumin (HSA) represents the most abundant plasma protein and accounts for 60–65% of the total plasma protein [8]. Along with maintaining the osmotic balance in the blood, this protein has multifunctional properties and influences many physiological parameters [9]. A key factor in the efficiency of albumin is its binding capacity as albumin binds toxins, such as uremic retention solutes and endotoxins as well as administered pharmaceutical agents efficiently [10,11]. The functions of albumin depend on changes in its 3D conformation [12]. Under pathological conditions, such as diabetes, renal and liver failure as well as critical illness, albumin binding is impaired, which can lead to a reduction in albumin’s function as a molecular scavenger [9]. 

Accumulating data indicate a link between albumin functionality and sepsis pathophysiology [13,14]. Therefore, the question arises whether the assessment of the albumin-binding function plays an essential role in sepsis diagnosis and prognosis. Albumin-binding capacity (ABiC) is a validated method to determine albumin-binding functionality, allowing insights into the binding site II specific loading state of the albumin molecule [15]. Several studies have already shown a correlation of ABiC with the clinical course of various diseases, e.g., liver and kidney failure [16,17]. 

Electron paramagnetic resonance spectroscopy (EPR)-based albumin function testing is another method for investigating the functionality of albumin regarding fatty acid binding [18,19,20]. The measurable spectra provide information about the transport-parameters-binding efficiency and detoxification efficiency. Initial pilot data showed significant differences in these transport parameters between patients who developed sepsis/systemic inflammatory response syndrome (SIRS) and those without such conditions [14]. The transport parameters also correlated with the courses of the patients’ diseases. Since the number of patients studied so far is too small, the available data do not yet allow any definitive conclusions to be drawn. It has already been shown that assessing albumin-binding function by a spin-labeled fatty acid technique shows a correlation with severity of liver disease [21,22] and is reduced in hemodialysis patients [23].

The aim of this study is to compare the diagnostic performance of two technically different methods, EPR and ABiC, both of which investigate specific functional parameters of albumin in intensive care. Specifically, the study aims to determine differences in binding and detoxification efficiencies between critically ill patients with proven sepsis and septic shock and a control group with no evidence of sepsis. 

## 2. Results

### 2.1. Study Population

Twenty-six patients were included in the present analysis. Of these, ten (38.5%) presented sepsis, and six (23.0%) presented septic shock at admission. Ten (38.5%) patients without evidence of positive sepsis criteria served as the reference population. Of all subjects, 18 (69.2%) were male. The median age was 59 years (range 32–96 years).

The patients presented with different reasons for intensive care unit admission. Table 1 provides further information. 

### 2.2. Albumin Binding in Patients on Intensive Care Unit (ICU) Admission

The binding capacity of albumin (ABiC) was determined using a dye-based method to quantify binding of a fluorescent probe at the binding site II of the albumin molecule. In parallel measurements on the same samples, the EPR technique using spin-labeled fatty acids was used to determine the following parameters: binding efficiency (BE); detoxification efficiency (DTE); binding coefficients at three different ethanol concentrations (KBA, KBB, KBC). Figure 1 shows the distribution of these parameters in the three patient groups. All patient samples were prepared for analysis once and measured in duplicate with both methods.

In patients suffering from sepsis or septic shock, ABiC, BE, and DTE are significantly reduced in comparison to the control group. A differentiation between sepsis and septic shock, however, was not possible. Binding coefficients at the lowest and middle ethanol concentrations (KBA and KBB) show significantly reduced values in patients with septic shock in comparison to controls but did not reach significance in patients with sepsis reduction. The binding coefficient at the highest ethanol concentrations (KBC) showed neither a reduction in patients with sepsis nor with septic shock in comparison to the controls. On the other hand, only KBC was significantly reduced in patients with septic shock in comparison to patients with sepsis.

Receiver operating characteristics (ROCs) of ABiC and BE are shown in Figure 2. This is a graphical plot that illustrates the diagnostic ability of a binary classifier system. We are aware that performing ROC curve analysis in a data set of n = 26 has only limited accuracy, but it allows a rough estimation of the data. With the areas under the receiver operating characteristic (AUROC) of 0.85 (0.70–0.94) and 0.86 (0.71–0.95) for ABiC and BE, respectively, both show equal diagnostic performances in differentiation of patients with sepsis from patients without sepsis. The same applies to the differentiation of patients with septic shock from control patients (AUROC: ABiC 0.89 (0.73–0.97), BE 0.89 (0.73–0.97)). A cut-off value of 70% ABiC would show a sensitivity of 85% at a specificity of 70% for control vs. sepsis and a sensitivity of 100% for control vs. septic shock. At a cut-off value of 38% BE, a sensitivity of 100% at a specificity of 70% would be reached for both control vs. sepsis as well as control vs. septic shock.

### 2.3. Comparison to Clinically Established Standard Parameters for the Assessment of the Patient’s Condition

A comparison with clinically established parameters for intensive care patients [1] was performed using the Spearman rank correlation. Therefore, albumin-binding parameters ABiC, BE, and DTE, were correlated to procalcitonin (PCT), C-reactive protein (CRP), and sequential organ failure assessment (SOFA) score (Table 2). Due to missing PCT values in three controls, a reduced dataset was analyzed (n = 23). Albumin-binding parameters show moderate correlation to PCT and SOFA score and weak correlation to CRP. ABiC did not reach significance for CRP.

### 2.4. Patient Survival

Patient survival was determined at 21 days after ICU admission. No patients died in the control group. Two of ten patients died in the sepsis group. Of these, one patient died after 2 days and the other after 18 days (median, IOR: 10 (6–14) days). The septic shock group showed higher mortality. A total of three of the six included patients died. The median survival time in this group amounted to 2 (2–5) days. Of note, one patient died after 99 days, i.e., outside the 21-day observation interval.

Six parameters were evaluated regarding their correlation with survival of the patients. Although none of them reached statistical significance, it should be noted that the total number of included patients in the study was limited (Table 3).

### 2.5. Multiple Linear Regression

Multiple linear regression analysis was performed to analyze the best joint predictors of ABiC, BE, and DTE, respectively. Four sets of variables were included, age and sex, inflammation-related parameters, such as PCT and CRP, liver-related parameters, such as bilirubin and albumin concentration, and kidney-related parameters, such as creatinine and urea.

Age and sex failed to show linear relationships for all three parameters. For BE and DTE, creatinine also showed no linear relationship. Analysis of the remaining parameters showed that PCT (*p* < 0.01), creatinine (*p* < 0.1), urea (*p* < 0.001), and bilirubin (*p* < 0.01) proved to be the best joint predictors of ABiC, explaining about two-thirds of the variance. For BE as well as for DTE, urea and CRP are the best joint predictors, explaining about half of the variance (BE: both *p* < 0.01; DTE: both *p* < 0.03). All the other variables did not reach significance.

### 2.6. Comparison of the Applied Methods for Albumin Binding Efficiency

It was possible to determine albumin-binding functionality and differentiate the control patients from the patients with sepsis or septic shock with equal performance. A direct comparison of ABiC and BE showed a linear relationship with a correlation coefficient of 0.81 (*p* < 0.001), whereas a comparison of ABiC and DTE showed a logarithmic correlation with a correlation coefficient of 0.86 (*p* < 0.001). 

Interestingly, the distribution of BE values appears to exhibit three distinct groups, which do not align with patient categorizations (Figure 3).

## 3. Discussion

The current pilot study aimed to gather essential preliminary data required for the design and implementation of a subsequent monitoring study involving patients in the field of intensive care medicine. This investigation focused on assessing the albumin binding and detoxification efficiency by utilizing an EPR spin-probe technique alongside the specific ABiC assay for determination of albumin’s binding site 2. The study encompassed intensive care patients afflicted with sepsis, septic shock, as well as a control group. Upon admission to the intensive care unit, it was observed that both the binding and detoxification efficiencies as well as the ABiC values exhibited statistically significant declines in patients diagnosed with sepsis or septic shock in comparison to the reference group.

Several studies have already investigated the effects of critical illnesses on albumin, focusing on its serum concentration [24,25,26]. In this regard, hypoalbuminemia is a rather common condition in critically ill patients [27] and has been linked to increased morbidity and mortality [26,28]. It is well known that albumin plays a pivotal role in controlling the body’s equilibrium of fluids [9], the transportation of various substances [29,30], and the regulation of inflammation [31,32] and oxidative stress [9,33,34]. However, the structural and functional integrity of the albumin molecule can be compromised by a series of influencing factors, including inflammation [31,32], oxidative stress [9,33,34], glycation [35], and diseases, such as liver disease [21,36]. These factors give rise to physiological alterations that do not only impact the concentration of serum albumin but also affect its binding and detoxification capabilities. In patients suffering from critical illnesses, the related dysfunction of albumin may aggravate the pathophysiological conditions and hamper the prognosis for patients of these conditions [14,25,31,37].

This hypothesis is also supported by our observations that functional parameters, which describe albumin’s binding capacity, such as ABiC, BE, and DTE, were reduced among patients suffering from sepsis or septic shock when compared to the control group. Furthermore, the inclusion of additional parameters, specifically KBB and KBC, allowed for discrimination between patients with sepsis and those suffering from septic shock. Our preliminary observations shown here underscore the potential for a meaningful characterization of the evaluated cohorts through the combinatory assessment of albumin functional parameters. As shown here, pathological conditions severely impact changes of albumin’s 3D conformation and consequently also the protein’s binding capacity.

Another elementary aspect of sepsis pathophysiology already appears in its definition, which assigns a special significance to the occurrence of organ insufficiencies [1,2].

Our observations align with correlations between various organ insufficiencies and the parameters under investigation. Multiple linear regression analysis revealed that PCT, creatinine, urea, and bilirubin collectively serve as the most reliable predictors of ABiC. In the case of the parameters, BE and DTE, CRP and urea were found to be the most robust common predictors. These findings offer valuable insights into the interplay between organ dysfunctions and both the functionality and performance of albumin. 

Of clinical significance, creatinine and urea have been established as surrogate markers of renal function, and their association with ABiC has been demonstrated in a previous study [17]. ABiC levels were reduced in patients with chronic renal failure, with a negative correlation observed between ABiC deterioration and the serum concentration of the uremic retention solute indoxyl sulfate—a toxin predominantly bound to albumin. ABiC was found to exhibit a strong correlation with the glomerular filtration rate [17]. Furthermore, Bilirubin is considered a clinically relevant indicator of liver function [38,39]. Our results on changes in the molecular conformation of albumin provide further evidence for a relationship between liver function in the context of sepsis. Several data support this hypothesis. For instance, analysis of albumin functionality using the EPR method in patients with diverse liver diseases demonstrated a significant reduction in detoxification efficiency (DTE) compared to the control group. Patients with acute-on-chronic liver failure (ACLF) exhibited an even greater decline in DTE compared to those with chronic liver disease but without ACLF. Additionally, DTE displayed a significant correlation with scores characterizing the severity of liver failure, such as the MELD score and the Child–Pugh score. [21,22]. Moreover, ABiC levels in patients with decompensated cirrhosis exhibited a negative correlation with disease severity as measured by the Child–Turcotte–Pugh (CTP) classification and the model for end-stage liver disease (MELD) [16,36].

Sepsis, a multifaceted syndrome characterized by a dysregulated host response to infection, is underpinned by complex pathomechanisms [1,2]. A central observation found in this pathophysiology is the aberrant production of reactive oxygen species (ROS). In the context of sepsis, these inflammation-related ROS play a pivotal role in orchestrating alterations in the conformation and functionality of albumin, primarily through oxidation of the critical Cys34 residue within the albumin molecule [32,33,34,40]. The analysis of binding coefficients and binding efficiency performed in our study, which provide clues about the molecular flexibility of the albumin molecule, suggested a detectable reduction in the molecular flexibility of albumin in patients with sepsis compared with the control group. This aligns with the observation of data indicating that in sepsis, the binding capacity of reversibly oxidized albumin known as human nonmercaptalbumin-1 (HNA-1) is impaired as determined by assessments using different techniques. [40,41]. 

In addition to the EPR method, Paar M. et al. utilizes another measurement method based on dansyl sarcosine, which evaluates fluorescence intensity and polarization of dansyl sarcosine bound to albumin in a solution, enabling the calculation of the binding constant and the number of binding sites through nonlinear regression analysis. In contrast, the ABiC method measures the fluorescence intensity of dansyl sarcosine that remains unbound after plasma sample filtration, determining the percentage of binding capacity using the above-mentioned formula. Therefore, various approaches exist to estimate albumin’s binding properties, even when utilizing the same ligand [17,41].

The results of the receiver operating characteristic (ROC) analyses further indicate the importance of albumin’s functional parameters by correlation with well-established markers of infection, namely procalcitonin (PCT) and the marker for inflammatory processes, namely C-reactive protein (CRP) levels [42]. Data from a recent observational study with a cohort of 87 ICU patients are consistent with these observations and provide further insight into the interplay between albumin and the inflammatory process mediated by reactive oxygen species (ROS) [31]. Moreover, Oettl and colleagues point out that albumin plays a central role in regulating the complex dynamics of inflammation and highlight its active interaction with ROS [40]. Their findings also suggest that albumin may play a significant role in the protein-dependent response to oxidative stress during inflammation, and hypoalbuminemia may result in a reduction in plasma antioxidant capacity.

A comparative analysis of both methods for the determination of albumin’s binding efficiency applied in our study revealed a strong correlation between ABiC, BE, and DTE. The disparities in the applied ligands may lead to variations in diagnostic performance depending on specific patient populations. ABiC employs dansyl sarcosine, which characterizes the binding capacity of albumin’s Sudlow side II, an area likely to be affected by disease-specific ligands, such as drugs or toxins [16,17,36,37]. On the other hand, EPR utilizes fatty acids known to interact with seven binding sites of albumin distributed across the protein, thereby encompassing Sudlow sides I and II as well [43]. Recently, Paar et al. introduced the above-mentioned dansyl-sarcosine-based method, which allows estimating binding constants and sites but some aspects inherent to the method limit its validity. It provides insights into albumin’s conformational changes through the fluorescence polarization measurement, which the ABiC test does not capture. However, Paar’s method may be susceptible to interference from other fluorescent substances in the solution, while the ABiC test effectively mitigates this issue by filtering the plasma sample. These distinct characteristics make each method valuable in studying albumin’s binding capacity with their respective advantages and limitations [17,41].

There are several techniques available for evaluating albumin’s binding properties, but to date, a direct method to measure albumin transport function remains unavailable. Thus, to facilitate the clinical application of our findings, a composite value, which combines the assessed parameters, could guide clinicians to monitor disease progression of patients in intensive care medicine. Although this pilot study offers only preliminary data, further investigations should be conducted within a controlled randomized setting. 

Our study has several strengths. To our knowledge, this is the first study to investigate the functional properties of albumin in critically ill patients with sepsis using two different methods and probes that interact specifically with the physiological sites. In this regard, spin labeling in combination with EPR spectroscopy as well as specific site II albumin-binding function assessed by the ABiC assay are helpful tools to study structural changes of albumin. Despite the valuable and promising results, this pilot study also has limitations. This is mostly due to the small number of patients who could be enrolled in this study. Therefore, the data have only limited statistical power, which becomes especially evident when analyzing the significance of the functional parameters in the disease subgroup (septic shock patients). Moreover, the small number of enrolled participants may be prone to selection bias or information bias, which would also influence the results. For example, the study excluded patients suffering from liver failure, which could potentially have a notable influence on the outcomes. To overcome these limitations, a new clinical study with a larger and also more diverse patient population will allow us to gain a more robust understanding of the implications of different albumin functionalities for sepsis patients.

Despite these recognized limitations, the study’s findings hold promise and may have a crucial impact on enhancing the diagnosis and prognosis assessment of sepsis patients. By acknowledging the potential limitations and considering these factors in future research endeavors, we can further build upon the valuable insights obtained from this pilot study. The observed results open avenues for future investigations with larger sample sizes and more comprehensive patient representations, allowing for a deeper exploration of the role of albumin in sepsis management. This, in turn, may lead to improved therapeutic approaches and better patient outcomes. Until then, cautious interpretation of the study’s findings is essential, while also recognizing the potential benefits they offer in advancing medical knowledge and potentially benefiting sepsis patients.

## 4. Material and Methods

### 4.1. Design, Settings and Study Population

The study population included patients enrolled in a prospective, observational study from October 2021 to February 2023 at the intensive care unit of the Department of Anesthesiology, Intensive Care Medicine and Pain Therapy and the Department for Internal Medicine, University Medical Center Rostock.

Inclusion criteria were (1) admission to the ICU, (2) diagnosis of sepsis according to the sepsis-3 definition (SSC), and (3) diagnosis of septic shock according to the sepsis-3 definition (SSC) at admission to the ICU [1,2].

Exclusion criteria were age < 18 years, extracorporeal treatment with citrate anticoagulation ongoing at the time of blood collection, human albumin administration for any reason, transfusion of more than 500 mL of fresh frozen plasma or erythrocyte concentrate 12 h prior to blood collection, liver failure (including liver cirrhosis), and chronic renal failure (CKD stage ≥ 4, GFR < 30 mL/min). A group of subjects without evidence of sepsis were analyzed as a control group.

This study was approved by the local ethics committee (Reg. No.: A 2020-0043). Written informed consent was obtained from patients or from legal representatives before enrollment, according to the 1975 Declaration of Helsinki. For each patient enrolled in the study, clinical and laboratory data, including hematological, coagulation, liver and renal parameters, electrolytes, serum CRP, and PCT were collected at admission. SOFA score was calculated to assess disease severity and prognosis. Finally, survival was recorded 21 days after ICU admission.

At the time of inclusion in the study, a peripheral blood sample was collected from a vein into EDTA tubes and lithium heparin tubes (Sarstedt, Nümbrecht, Germany).

Blood samples were centrifuged at 3000× *g* for 10 min; plasma was aliquoted into cryotubes (Corning Inc., Corning BV, Amsterdam, The Netherlands) and stored at −80 °C until analysis. All patient samples were prepared for analysis once and measured in duplicate with both methods.

### 4.2. Measurement of Albumin Concentration

Determination of albumin concentration is a prerequisite for determination of ABiC. A colorimetric assay was used to measure albumin concentration. Measurements were performed on the CobasR Mira Plus automated analyzer (Hoffmann-La Roche AG, Basel, Switzerland) using the reagent “ALBUMIN Bromcresol Green” (Labor + Technik Eberhard Lehmann GmbH, Berlin, Germany).

### 4.3. Estimation of ABiC

ABiC was determined by an indirect method based on the estimation of the unbound fraction of a specific albumin-bound marker in a plasma sample. By comparison with the fraction of the unbound marker in a reference albumin solution, the site-specific binding capacity of the sample was expressed semiquantitatively. Plasma samples were diluted to an albumin concentration of 150 mmol/L and incubated with an albumin-binding site II -specific fluorescent marker [dansyl sarcosine (DS), 150 mmol/L]. Albumin-free filtrate was obtained in a separation step (Centrisart I, 20,000D; Sartorius GmbH, Göttingen, Germany), and fluorescence was measured after addition of human serum albumin (300 mmol/L) as a fluorescence enhancer (Fluoroscan 355/465 nm; Labsystems Diagnostics Oy, Vantaa, Finland). In parallel, the same procedure was performed with standard albumin as reference. A standardized virus-inactivated human serum preparation from pooled human plasma (Biseko^®^; Biotest Pharma GmbH, Dreieich, Germany) was used as a reference for ABiC. The binding capacity for the marker was quantified according to the following equation:$$ABiC\;\left[\%\right]=\frac{fluorescence\;in\;ultrafiltrate\;of\;the\;reference\;\;{Biseko}^{®}}{fluorescence\;in\;ultrafiltrate\;of\;the\;sample}\times 100$$

### 4.4. Determination of Binding and Detoxification Efficiency by Electron Paramagnetic Resonance (EPR)-Spectroscopy

All samples were blinded for the clinical data and sent to MedInnovation GmbH (Berlin, Germany) for assessing albumin-binding function by the electron paramagnetic resonance (EPR) spin-probe technique, using a commercially available EPR spectrometer (EPR-Analyzer, MedInnovation GmbH, Berlin, Germany). Each sample was measured with an EPR spin-probe technique as described [19] and as briefly below: 

Characterization of spin-probe-binding affinity to albumin was realized by incubation with varying ethanol concentrations and changes in the relationship of spin probe to albumin concentration at different hydrophobic conditions. Commercial 16-doxyl stearic acid (Avanti Polar Lipids Inc., Alabaster, AL, USA) was applied as a spin probe on the basis of an extremely high binding constant (6.9 × 10^7^ L/mol) for albumin to this stearic acid, generally leading to >99.9% binding of this spin probe to albumin [44]. Modification in binding affinity was induced by administration of an extra amount of pure ethanol (Carl Roth GmbH & Co.KG, Karlsruhe, Germany). Time from thawing to analysis of the frozen serum samples was within 30 min. Three separate aliquots were prepared from each sample. To prepare each aliquot, 50 µL of serum were mixed in a microtiter plate well with a specified amount of an ethanol-containing spin probe to achieve aliquots differing in the concentration of spin probe and ethanol (aliquot A—17 Vol% ethanol and fatty acid/albumin ratio 1:1, aliquot B—19 Vol% ethanol and FA/alb ratio 2:1, aliquot C—22 Vol% ethanol and FA/alb ratio 3:1). Different concentrations of ethanol are used to induce an opening of the albumin molecule to a different extent to simulate loading, transport, and unloading conditions [45,46]. With increasing ethanol concentrations, the binding affinity of albumin for 16-doxyl stearic acid is decreasing, simulating the release of ligands. Thus, the conformational flexibility of the albumin molecule can be analyzed. The serum/ethanol mixtures were covered with Parafilm and then incubated for 10 min at 37 °C on a microplate shaker. Afterwards, aliquots were brought into capillary glass tubes for analysis within the EPR-Analyzer (temperature: 37 °C). The EPR-Analyzer utilized a microwave power of 15 mW at a frequency of 9.4 GHz. The magnetic field strength with a scan range of 12 mT was 0.34 T, embedding a modulation amplitude of 0.07 mT. EPR spectra were examined with specific software (version 4.0.1 developed by MedInnovation GmbH). Analysis of the EPR spectra followed a complex mathematical computer simulation of its components related to a Hamilton spin function with axial anisotropy also described in detail previously [47]. From that simulation, relative spin-probe concentrations in different components are calculated. Binding and detoxification efficiency (BE and DTE) were calculated as regression of the relative spin-probe concentrations, C1 and C2, over the three ethanol concentrations (Patent WO2021038032; PCT/EP2001/002248). C1 is the relative spin-probe concentration of the EPR spectrum of the spin probe bound to the primary binding site, where 16-doxyl stearic acid is strongly immobilized and C2 of the secondary binding site of albumin where 16-doxyl stearic acid is weakly immobilized [20]. BE determines the strength and amount of bound fatty acids under the certain conditions of ethanol concentrations, while DTE determines the molecular flexibility of the patient’s albumin; thus, the ability to change the conformation depends on the ethanol concentration (Table 4) [21,48]. Both are normalized to published data of healthy individuals (349 blood donors, median age 38 years, 54% male) and are expressed in % [19]. Binding coefficients determine the ability of binding fatty acids under the specific ethanol concentrations, with KBA at the lowest, KBB at the middle, and KBC at the highest ethanol concentration. Binding coefficients are expressed in M^−1^. 

### 4.5. Statistical Analysis

Statistical analysis of the data was performed using IBM SPSS Statistics (version 27, Chicago, IL, USA) and GraphPad Prism (version 9, GraphPad Software Inc., San Diego, CA, USA). Results are expressed as median ± 95% confidence interval. Violin plots, dot plots, and ROC curves were used for graphics. Samples were measured in duplicates. All statistical analyses were performed with mean of duplicates.

According to the distribution of data (using the Shapiro–Wilk test), the Mann–Whitney test was used for two independent samples for continuous variables. The Kruskal–Wallis test was used to test differences between multiple independent samples with non-normal underlying population distribution, and appropriate post hoc tests were applied if necessary (Tukey’s multiple comparisons test). Statistical differences were considered significant at *p*-value < 0.05 and highly significant at *p*-value < 0.01.

Correlation analysis was performed using the Spearman rank correlation. Multiple linear regression analysis (MedCalC^®^ version 11.4.2.0) was used to find a subset of independent variables (sex, age, PCT, CRP, creatinine, urea, bilirubin, and albumin) being the best joint predictors of ABiC, BE, and DTE, respectively, which each were used as dependent variable. Sex was coded as 1 (women) or 2 (men); all other variables were used as continuous variables. A backward analysis with *p* < 0.05 to enter and *p* > 0.1 to remove variables was used. Linear relationships of each independent variable with ABiC, BE, and DTE, respectively, were prerequisites for multiple linear regression analysis. *p*-values < 0.05 were considered to be significant.

## 5. Conclusions

The incorporation of EPR spectroscopic measurements in tandem with the ABiC assay to evaluate albumin-binding efficiency, alongside the utilization of well-established biochemical markers, such as procalcitonin, C-reactive protein, or cytokins, exemplifies an inventive approach. This integration holds considerable promise in substantially enhancing the assessment and monitoring capacities for critically ill patients. Nonetheless, attaining a more comprehensive understanding of the underlying pathophysiology and disease progression remains imperative to enhance clinical decision-making and optimize patient outcomes in these intricate clinical scenarios.

## Figures and Tables

**Figure 1 ijms-24-12551-f001:**
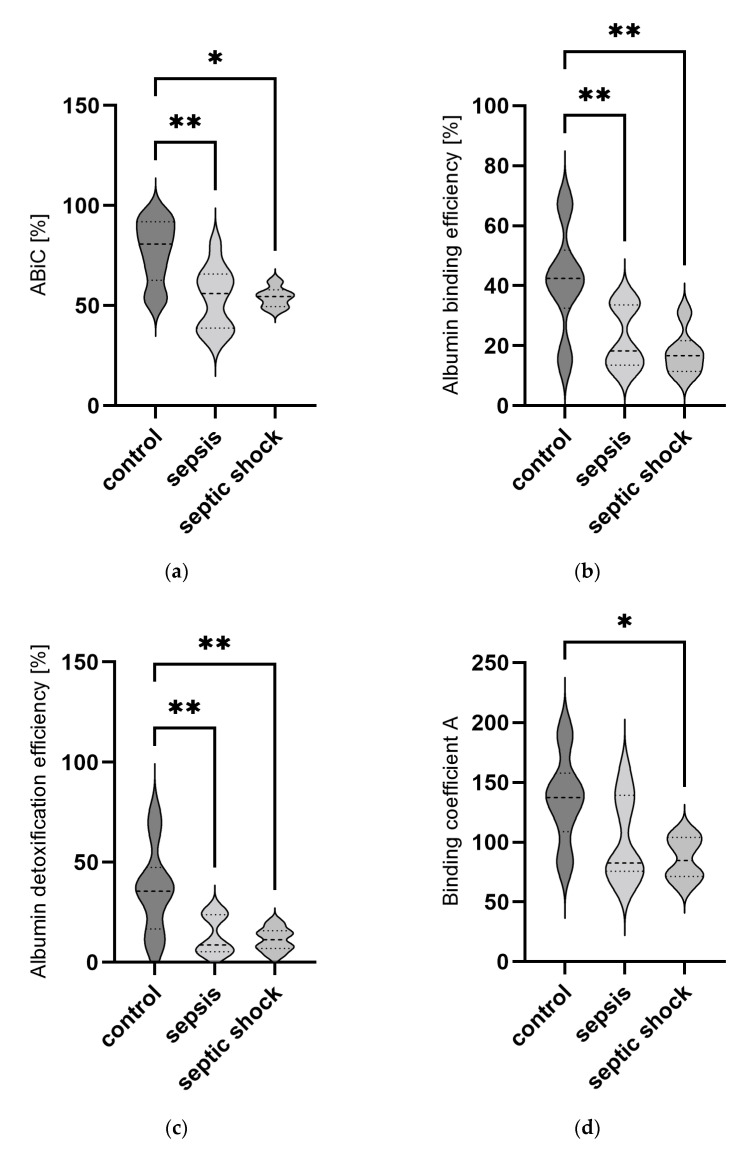
Violin plot of (**a**) ABiC, (**b**) BE, (**c**) DTE, (**d**) KBA, (**e**) KBB, and (**f**) KBC in ICU patients without sepsis or septic shock (control, n = 10), patients with sepsis (n = 10) and patients with septic shock (n = 6) at admission. All patient samples were prepared for analysis once and measured in duplicate with both methods. ABiC, BE, and DTE are expressed in %. KBA, KBB, and KBC are expressed in M^−1^. The horizontal line within the violin plots represents the median, whereas the upper part represents the 75th and the lower part the 25th percentiles. *p* values ≤ 0.05 (*) and ≤ 0.01 (**) were considered significant.

**Figure 2 ijms-24-12551-f002:**
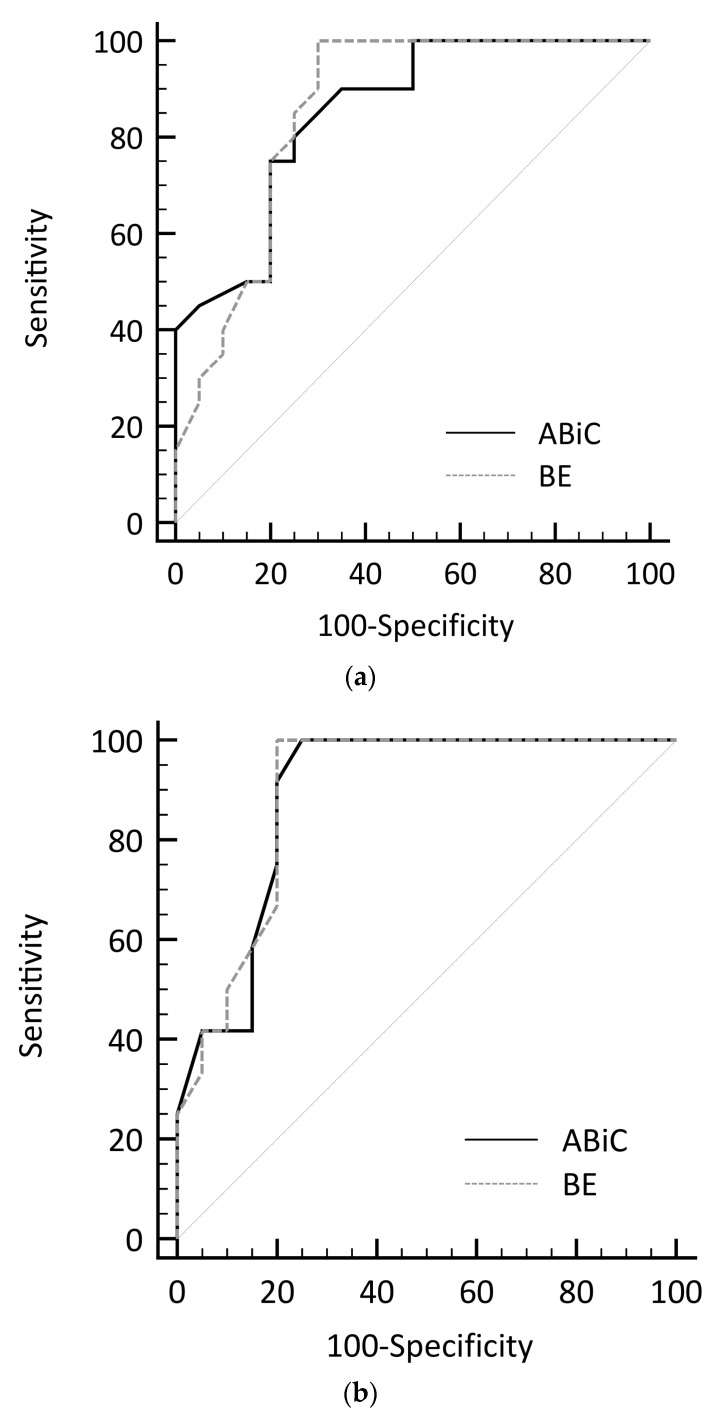
Receiver operating characteristic curves of ABiC and BE for controls vs. patients with sepsis (**a**) and controls vs. patients with septic shock (**b**). (ABiC—albumin-binding capacity, BE—binding efficiency). The diagonal line on the graph indicates points where sensitivity equals 1 minus specificity. A test with points falling on this line is entirely ineffective and lacks diagnostic value.

**Figure 3 ijms-24-12551-f003:**
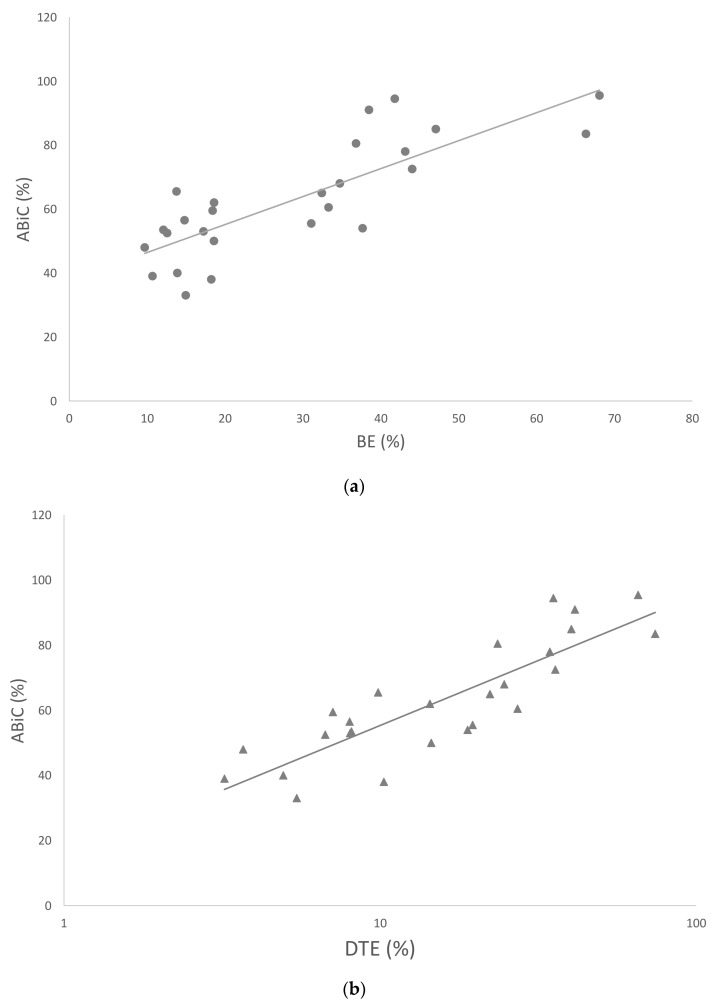
Spearman rank correlation analysis of n = 26 samples shows a linear relationship between ABiC and BE (**a ●**) and logarithmic correlation between ABiC and DTE (**b ▲**). The straight line represents regression line.

**Table 1 ijms-24-12551-t001:** Clinical Data of Patients at Intensive Care Unit (ICU) Admission.

	Control (*n* = 10)	Sepsis (*n* = 10)	Septic Shock (*n* = 6)	*p* Value
Anthropometric data				
Age (years)	58.5 (49.3–62.0)	62.5 (52.8–79.0)	61 (54.8–68.0)	0.551
Male sex	6 (60)	7 (70)	5 (83.3)	
Hematology, biochemistry, and prognostic scores				
Hemoglobin (mmol/L)	6.9 (5.9–8.7)	4.9 (4.5–5.5)	5.9 (5.4–6.2)	0.141
Erythrocytes (10^12^/L)	3.5 (2.8–3.9)	2.7 (2.4–2.8)	2.9 (2.8–3.2)	0.026
Hematocrit	0.31 (0.26–0.33)	0.26 (0.23–0.26)	0.28 (0.26–0.30)	0.005
Platelets (10^9^/L)	227 (211–266)	128 (86–272)	114 (62–180)	0.052
Leukocytes (10^9^/L)	9.6 (7.1–11.6)	10.2 (8.2–12.2)	15.5 (13.0–17.0)	0.382
C-reactive protein (CRP) (mg/L)	85 (56–120)	151 (126–206)	161 (122–242)	0.037
Procalcitonin (PCT) (ng/mL)	0.1 (0.1–0.2)	1.1 (0.6–3.0)	11.5 (7.8–18.6)	<0.0001
Lactate (mg/dL)	9.0 (5.8–13.3)	4.6 (0.7–10.6)	21.6 (18.5–44.4)	0.02
Albumin (g/L)	36 (28.7–38.1)	22.5 (20.9–30.2)	20.7 (19.7–22.1)	0.005
Bilirubin (µmol/L)	8.5 (3.8–12.5)	8.5 (6.0–11.8)	30.0 (27.0–38.0)	<0.0001
Creatinine (µmol/L)	57.5 (54.3–75.5)	79.0 (60.8–100.5)	108.5 (83.3–149.5)	0.107
Urea (mmol/L)	5.5 (4.6–6.8)	8.6 (5.7–17.3)	14.6 (9.5–21.9)	0.029
Sodium (mmol/L)	139 (138–140)	139 (136–140)	134 (131–137)	0.162
Potassium (mmol/L)	4.0 (3.8–4.2)	3.9 (3.6–4.3)	4.7 (4.4–4.9)	0.002
Calcium (mmol/L)	1.2 (1.2–1.2)	1.1 (1.1–1.2)	1.2 (1.1–1.3)	0.192
Sequential organ failure assessment (SOFA) score	1.0 (0.3–1.0)	6.0 (4.3–7.5)	10 (10.0–11.5)	<0.0001
Reasons for ICU admission n (%)				
Hospital-aquired pneumonia	0 (0)	5 (50)	2 (33.3)	
Soft tissue infection	0 (0)	2 (20)	1 (16.6)	
Urosepsis	0 (0)	1 (10)	0 (0)	
Abdominal infection	0 (0)	1 (10)	0 (0)	
Spondylodiscitis	0 (0)	1 (10)	0 (0)	
Mediastinitis	0 (0)	0 (0)	2 (33.3)	
Endocarditis	0 (0)	0 (0)	1 (16.6)	

**Table 2 ijms-24-12551-t002:** Spearman rank correlation analysis (Spearman’s coefficient of rank correlation (rho)).

	ABiC (%)		BE (%)		DTE (%)	
Parameter	r	*p*-Value	r	*p*-Value	r	*p*-Value
PCT (ng/mL)	−0.55	0.0064	−0.73	0.0001	−0.62	0.0016
CRP (mg/L)	−0.39	0.0656	−0.48	0.0219	−0.42	0.0455
SOFA score	−0.53	0.0098	−0.64	0.0010	−0.56	0.0058

**Table 3 ijms-24-12551-t003:** Comparison of survivors and non-survivors. Data are expressed as median (IQR) and *p*-values.

Parameter	Survivors (n = 21)	Non-Survivors (n = 5)	*p*-Value
ABiC (%)	62 (52–81)	54 (52–62)	0.313
BE (%)	31 (15–42)	19 (12–33)	0.255
DTE (%)	19 (8–35)	15 (8–24)	0.720
PCT (ng/mL)	1.0 (0.1–3.5)	4.4 (0.5–15)	0.157
CRP (mg/L)	124 (61–189)	148 (120–213)	0.283
SOFA score	4 (1–8)	8 (4–12)	0.090

**Table 4 ijms-24-12551-t004:** EPR binding properties.

Parameter	Description
BE (kb_G_ × n_G_)	binding efficiency
DTE (LQ/UQ)	detoxification efficiency
KB A [(1 − C3)/(C_Alb_ × C3)]	binding coefficient of albumin at highest ethanol concentration (17 Vol%)
KB B [(1 − C3)/(C_Alb_ × C3)]	binding coefficient of albumin at highest ethanol concentration (19 Vol%)
KB C [(1 − C3)/(C_Alb_ × C3)]	binding coefficient of albumin at highest ethanol concentration (22 Vol%)
kb_G_	global binding constant of whole albumin molecule, determined as regression of ratios between relative amounts of spin-probe molecules within both binding sides, strongly and weakly immobilized, to the concentration of ethanol
n_G_	amount of spin-probe molecules within both binding sides, strongly and weakly immobilized
LQ	loading quality—binding property under low ethanol (17 Vol%) and low spin probe (spin probe/albumin ratio 1) conditions
UQ	unloading quality—binding property under high ethanol (22 Vol%) and high spin probe (spin probe/albumin ratio 3) conditions
C3	fraction of unbound fatty acids
C_Alb_	patient’s albumin concentration

## Data Availability

The datasets used and analyzed during the current study are available from the corresponding authors on reasonable request.

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
