# Peer review of "Exploring Albumin Functionality Assays: A Pilot Study on Sepsis Evaluation in Intensive Care Medicine"

_ijms, 2023, doi:10.3390/ijms241612551_

Round 1
Reviewer 1 Report
The manuscript by Klinkmann et al. aims to determine albumin binding function in patients with sepsis or septic shock applying two different methods. With the resulting parameters it is possible to distinguish between control patients and patients with sepsis or septic shock. The parameters determined with the two different methods (using different ligands) correlate with each other. Although the topic is quite interesting I have several concerns about the manuscript and cannot suggest its publication in the present form. The discussion section is overstretched and, at many parts, lacks appropriate references. But most importantly, the statistics are doubtful. Duplicate data points are presented in each figure and obviously were used for calculation of differences within the groups to achieve statistical significance. This is not good scientific practice.
Further comments:
1.) Abstract
line 16: “binder of toxins” needs rephrasing
line 30: explain the abbreviation PCT
2.) Introduction
lines 40-42: information is missing whether the stated numbers refer to worldwide cases
line 47: citation must be added appropriately (Evans?)
line 50: how can albumin bind “solutions”? maybe solutes are meant?
lines 56-57: sentence needs rephrasing; as the authors reported, albumin has several functions and not only one, thus, what is meant by “quantification of albumin function”?
line 59: “site-specific loading state” - what is that supposed to be?
line 67: explain abbreviation SIRS
lines 73-74: objective needs rephrasing; assessment of “reliability” of a method which has been used in many studies before is devastating and questions previous results.
line 76: which different patient groups? in this study only sepsis patients are described
3.) Results:
characterization of control and patient groups is missing; which diseases brought them to the ICU? were there patients with liver disease among the septic shock patients?
Table 1: Erythrocyte and Leukocyte should be expressed in plural; explain the abbreviation SOFA
lines 91-92, Figure 1: duplicates are depicted as distinct data points, which deludes that the sample size of each group was bigger than in reality. How were the statistics calculated- also with duplicate data? this must not be - make a mean of the duplicates or show a representative experiment of two.
Figure 1, legend: M-1 is not a correct unit
lines 122-125: this sentence needs rephrasing
Table 2: number of study subjects (14/20/12) is not the same as at the beginning of the manuscript (10/10/6)
Table 3: survivors + nonsurvivors = 24; what happened to the missing two patients?
line 151: I don’t understand that; please rephrase
lines 153-157: switch paragraph to the discussion section; it also needs a reference
Figure 3: measured twice – again duplicates?
lines 168-170: Spearman or Pearson? I couldn’t find a description about bivariate correlations in the material and methods section
Figure 4: only independent data points should be presented, no duplicates
Figure 4, upper panel: the distribution of the data looks like different groups, I wouldn’t put them all into the same pot but analyze the groups separately
4.) Discussion:
line 180: “control population” is not a correct description for the 10 control patients from ICU investigated in this study
line 190-192: add references; #31 is only about glycation
line 219: how many patients suffered liver failure?
line 235-238: add appropriate references for this sentence; reference #30 doesn’t explain that.
lines 241-244: explain abbreviation HNA-1; no data about HNA-1 in sepsis patients are presented in the manuscript
line 248-253: reference #29 reports only about HSA concentration, no ROS
line 265: KBC/KBA-ratio needs a reference
lines 266-271: I cannot find the information about how the mentioned parameters were combined with BE and DTE within the cited references. Please explain or rephrase.
line 286: it was only one physiological ligand
line 289-298: devastating! needs to be rephrased
5.) Material and Methods
The study population needs to be described in more detail. Which underlying diseases brought them to the ICU? E.g., liver cirrhosis was an exclusion criterion – but what about other liver diseases?
line 325: were there two independent experiments performed or only one with samples in duplicates?
lines 364-365: >99.9% binding – please specify at which concentration range this high binding affinity can be expected; add reference
line 386: please explain in detail, how BE and DTE are calculated as “regression” of the relative spin probe concentrations?
lines 387-390: reference #16 alone does not explain this; please add appropriate structure biology data or references showing that conformation and flexibility of albumin are altered in the presence of ethanol in patients
line 394: M-1 is a strange unit for a binding coefficient
lines 395-397: this sentence should be removed; the binding of ligands to albumin is a complex scenario, harboring many different binding sites for a variety of ligands with different affinities; with the EPR method applied in this study only fatty acid binding is measured and even at the highest fatty acid concentration used, not all fatty acid binding sites will be occupied; a generalization for other ligands is therefore not convincing.
line 407: which post hoc tests were applied?
add information about the calculation of correlation coefficients (Figure 4) to the Statistical analysis section
Minor editing of English language required
Author Response
Dear Reviewer, we are grateful for your detailed analysis of our manuscript. Please find below our responses to your comments. Please let us first comment on the general remarks. As requested, we have revised the presentation of the methodology extensively and presented our approach in detail and transparently. The discussion has been further refined, supported with relevant sources and shortened as requested. In addition, the presentation of the limitations of our statements was again strongly prioritized.
The statistical analysis has been carried out according to scientific standards, as described in the methods section. In particular, we would like to point out that no significance test was performed with duplicated data points, but with the mean values of corresponding measurement series. The graphical representation, however, was done with the duplicated data points. This obviously conveyed a non-ambiguous message. Therefore, we have explicitly pointed this out again here and also in the text and adjusted the illustrations.
We honestly hope that this response to your comments is satisfactory. We believe that the signals provided by our study may serve as an encouraging basis for further studies and may contribute to the scientific community's understanding of the importance of the albumin molecule.

Reviewer 2 Report
Manuscript title: “Exploring Albumin Functionality Assays: A pilot study on sepsis evaluation in intensive care medicine”
Authors: Gerd Klinkmann et al.
Submitted to IJMS
The manuscript describes the use of a novel EPR-bases assay to investigate sepsis related blood values for diagnostics. Blood samples from patients in intensive care are compared for sepsis and non-sepsis cases and the albumin properties from their blood are compared. The finding is that the novel EPR assay has the potential to be a diagnostic tool for the future, given better differentiation between conditions than the previously used tests.
This clearly is an important contribution and therefore merits publication, even though as a non-specialist in the field the meaning of the statistical findings and how they compare to current standards is outside my field of expertise. From the EPR point of view, the results certainly are plausible and the approach looks sound.
Of course, the question if a clinical application is useful/costefficient etc. is beyond my scope of expertise. With the exception of the points mentioned below, the manuscript is written clearly and suitable for publication in IJMS.
The manuscript could be improved by considering the following points:
1. Line 89, p3:
„ as binding coefficients A, B and C (BE, DTE, KBA, KBB, KBC)“
the meaning of these coefficients and the reference should be given at this point and not at the end of the manuscript
furthermore, for the non-clinical audience the term
“receiver operating characteristic curves” should be explained.
Author Response
Thank you very much for your analysis of our manuscript. Please find the desired changes in the text.
Reviewer 3 Report
In this work the Authors discuss the reliability of the EPR method in evaluating the transport properties of albumin and compare it with the ABiC method. Furthermore, they assessed if these tools could be used as biomarkers for sepsis/septic shock diagnosis and prognosis in ICU patients. The concept of finding always new biomarkers able to help physicians in facing this challenging clinical entities is really stimulating.
Overall, this paper provides different interesting aspects. Anyway, there are different concerns that should be addressed:
1. Abstract:
- Abstract is too long. The Instruction for Authors guidelines explicit “The abstract should be a total of about 200 words maximum”. Please, shorten this part accordingly.
- Furthermore, there are different typos and word repetitions which lead this part redundant.
- Lines 19-20: I do not really understand this sentence. I humbly think that it should be rephrased.
2. Introduction:
- The first part of this section gives a good-quality introduction to the concepts of sepsis and septic shock. However, after line 45, I suggest to slightly expand the problem of mortality. Why the mortality rate is so high (e.g. challenging recognition, comorbidities, complexity of treatments)? Since this latter issue (complexity of treatments) is a very important point, I suggest to add a small paragraph about it and include the following reference: Guarino M, Perna B, Cesaro AE, Maritati M, Spampinato MD, Contini C, De Giorgio R. 2023 Update on Sepsis and Septic Shock in Adult Patients: Management in the Emergency Department. J Clin Med. 2023; 12:3188.
- Line 47: what does Evans between the brackets mean?
- Even in this section there are some typos (e.g. double spaces or points before and after brackets) which should be revised.
3. Results:
- Lines 88-89: the abbreviation used are not really clear. Please, better specify them (in particular “which one is directed to what”) in order to clarify this part and make the rest of this section easier to read.
- Lines 107-109: The authors stated “With AUROC of 0.85 (0.70-0.94) and 0.86 (0.71-0.95) for ABiC and BE, respectively, both show equal diagnostic performance in differentiation of patients with sepsis from patients without”. I have different concerns about this sentence. Actually, I agree that there is an important difference between ABiC/BE in septic vs. non-septic patients. However, as stated above, ABiC might correlate with patients’ prognosis also in liver/renal failure. Have you excluded these possible confounding factors in your analysis?
- Paragraph 2.4: The Authors assessed 21-day survival as outcome. Have you tried to assess shorter term outcome (e.g. 7-day mortality). As known, hospitalization (in particular ICU-stay) might be burdened by different complications which can confound your results.
- Paragraph 2.5: This part is quite interesting even if the results seem to exceed the main outcome of the analysis. Might you highlight these findings in a table to make them easier to be read.
4. Discussion:
- I suggest to start the discussion highlighting the main findings of your research without repeat the aims of the study again.
- Line 194, 214 and 245: you used the word “Notably”. I humbly think that if a sentence is stated in a scientific manuscript is itself notable, thus leading this word redundant. Please delete.
- Line 208-209: The Authors stated that “Multiple linear regression analysis revealed that PCT, creatinine, urea, and bilirubin collectively serve as the most reliable predictors of ABiC”. Although I agree with this concept, I can not highlight that both creatinine and bilirubin are part of the SOFA score. How this might influence the result? A methodologic well-performed multivariate analysis should not consider parameters which are already in the definition of the included population (i.e. septic patients). Please comment.
- Lines 232-244: Although surely interesting, this part is too long. Please shorten it a little to make this part easier to be read.
- Even in this section there are several typos that should be fixed.
5. Materials and methods:
- Paragraph 4.5: I think that you included too many variables in the multivariate analysis. For the sake of clarity, a correct analysis should include one variable for every ten records. In your case, you should perform it with three variables at most.
- Even in this section there are several typos that should be fixed.
6. Conclusion:
- The Authors should avoid to start conclusion explaining again how do they obtain the findings reported. This will make this part easier or the reader.
Moderate editing of English language is required
Author Response

(The authors gave the same response as above.)

Round 2
Reviewer 1 Report
The manuscript ‘Exploring Albumin Functionality Assays: A pilot study on sepsis evaluation in intensive care medicine’ has been thoroughly revised by the authors and most ambiguities have been clarified. The results are now clearly presented, appropriate references were added, and also the statistics are fine.
Minor comments:
lines 225 – 226: the levels of bilirubin were quite high, especially in the septic shock group. Can the authors explain where these come from, given, that liver failure was an exclusion criterion?
lines 287 – 289: This sentence could be slightly rephrased into a future perspective.
Author Response
We are grateful to you for the critical discussion of our manuscript. We would like to comment on your question below:
Bilirubin is a breakdown product of hemoglobin, which is typically detoxified by the liver and excreted with bile. In the case of septic shock, liver function may be compromised, leading to reduced bilirubin excretion. [Wang D, Yin Y, Yao Y. Advances in sepsis-associated liver dysfunction. Burns Trauma. 2014 Jul 28;2(3):97-105. doi: 10.4103/2321-3868.132689.]
Furthermore, you will find a rephrased version of the sentence in the text.
Yours sincerely
Gerd Klinkmann
Reviewer 3 Report
Dear Authors,
almost every comment has been amended. Therefore, the manuscript now is suitable for publication.
Sincerely.
Author Response
We would like to express our sincere thanks for your comments and the positive evaluation of our manuscript.
Yours sincerely
Gerd Klinkmann